# The Impacts of Shopping Tourism on Retail Sales and Rents: Lessons from the COVID-19 Quasi-Experiment of Hong Kong

Chung Yim Yiu

Department of Property, Business School, The University of Auckland, Auckland 1010, New Zealand;
edward.yiu@auckland.ac.nz

**Abstract:** This research studies the impact of shopping tourism on retail sales and rents, using the COVID-19 pandemic as a quasi-experiment. Shopping tourism refers to individuals who travel primarily for shopping purposes, and their spending patterns can have significant effects on the retail sector. The COVID-19 pandemic disrupted global travel and resulted in a decline in shopping tourist arrivals, leading to a downturn in sales for retailers dependent on shopping tourism. Additionally, the decline in shopping tourism affected retail rents, as the reduced demand for retail spaces posed challenges for property owners in attracting tenants. The study focuses on Hong Kong, a prominent shopping destination, which experienced a significant decline in tourist arrivals and retail sales during the pandemic. The research analyses the relationship between tourist arrivals, retail sales, and rents using time series analysis and identifies the impact of shopping tourism on retail rents. The findings suggest a positive association between tourist arrivals and retail sales and rents, particularly during the period of shopping tourism growth. However, the pandemic severely reduced this effect, revealing the impact of shopping tourists on the retail sector. The study concludes by discussing the implications for retail resilience and highlighting the need for further research on the impacts of shopping tourism on retail sales and rents.

**Keywords:** retail sales; retail rents; shopping tourism; COVID-19; retail resilience; Hong Kong

## 1. Introduction

Tourism has long been a catalyst for economic growth and development, with retail shopping playing a significant role in attracting visitors and boosting local economies. With the booming of shopping tourism in recent decades, retail markets in some popular shopping tourist destinations have become highly dependent on tourism. However, the outbreak of the COVID-19 pandemic disrupted global travel patterns, providing a unique opportunity to examine the impacts of shopping tourism on retail sales and retail rents. By exploring the quasi-experiment presented by the pandemic, we can gain valuable insights into the relationship between tourism, retail, and economic resilience.

"Shopping tourists" are individuals who travel with the primary intention of engaging in shopping activities. They seek destinations known for their shopping opportunities, whether for unique products, luxury goods, favourable prices, tax exemptions, or exclusive shopping experiences. These tourists dedicate a significant portion of their travel itinerary to shopping, and their spending patterns can have profound effects on the retail sector.

The COVID-19 pandemic significantly disrupted shopping tourism, leading to a drastic decline in shopping tourist arrivals worldwide. The restrictions on international travel and lockdown measures implemented to curb the spread of the virus resulted in a substantial reduction in the number of shopping tourists and their spending capacity. As a consequence, retailers that heavily relied on shopping tourism experienced a significant downturn in sales.

The decline in shopping tourism had implications for retail rents as well. Prior to the pandemic, shopping tourist destinations often witnessed high demand for retail spaces,

leading to increased rental prices. Property owners and landlords capitalised on the popularity of these areas, attracting tenants who catered specifically to shopping tourists. However, the pandemic-induced decline in shopping tourism reduced the demand for retail spaces, creating challenges for property owners in attracting tenants and maintaining rental income. There have been very few empirical studies on the impacts of shopping tourism on retail sales and rents, especially ones using exogenous shocks as a quasi-experiment to mitigate endogeneity biases.

This study takes Hong Kong as the city for studying the impact of shopping tourism on retail sales and rents, as its retail markets were one of the hardest hit markets during the pandemic. The year-on-year change in retail sales in Q1 2020 in Hong Kong saw a record low of −35.0%. The depth of the plummet in Hong Kong was one of the most significant in the world and was almost seven times of that in the USA.

Hong Kong has long been recognized as a prominent shopping destination, attracting millions of shopping tourists each year. Its reputation as a shopping paradise was built on a combination of factors, including a wide range of retail options, competitive prices, and a favourable tax environment.

One of the key drivers of shopping tourism in Hong Kong is the presence of luxury brand stores and shopping malls offering high-end fashion, jewellery, and electronics. Tourists, particularly from Mainland China and neighbouring Asian countries, flock to Hong Kong to take advantage of the diverse shopping opportunities and purchase luxury goods, often at lower prices and lower taxes compared to their home countries.

### 1.1. Tourism and Retail Sales of Hong Kong

Tourism is one of the four key industries in Hong Kong, with inbound tourism contributing 3.6% of the GDP in 2018. The value added by the inbound tourism industry increased from HK$37 billion in 2008 to HK$98 billion in 2018, representing an average annual growth rate of 10.3%. The industry employs 5.8% of the labour population. However, its contribution to GDP fell to 0.1% in 2021 (HK C&SD 2022).

Before the pandemic, the growth of the number of tourists in Hong Kong was spectacular. The increase in the number of tourists, by 523% (from 988,705 to 6,155,269 per month), substantially increased the demand for retail shops and was reflected in the 125% increase in retail sales (in real terms) from Q1 2000 to Q4 2018. However, the association between the two was not strong before 2009, when shopping tourism was not common. The implementation of the Individual Visit Scheme (IVS) in 2003 and the Multiple-Entry Permit (MEP) scheme in 2009, have led to an increasing trend of cross-border shopping tourism in Hong Kong (Li et al. 2018). IVS is a scheme that allows residents from a large number of cities of Mainland China to visit Hong Kong as individuals, and MEP is a scheme that allows eligible residents in Shenzhen to have a multiple-entry permit to travel freely to Hong Kong for leisure and shopping activities.

Figure 1a,b shows the scatterplots of the year-on-year changes of retail sales amounts and number of tourists in the periods 1999–2008 and 2009–2019. The former shows a weak association between the two, as shopping tourism was not popular, whereas the latter shows a much stronger positive association between the two; during this period, same-day visitors, who are mostly shopping tourists, accounted for more than half of the total number of tourists.

However, in the wake of the COVID-19 pandemic, there was a dramatic and unparalleled decline in tourist arrivals. The viral outbreak was officially declared by the Hong Kong government on 25 January 2020, leading to the suspension of high-speed rail services and most cross-border ferry services on 30 January. Subsequently, on 25 March, Hong Kong closed its borders to all non-residents. As a result, the number of tourists in Q2 2020 experienced a staggering 99.9% year-on-year decrease. The lowest recorded number of tourist arrivals occurred in Q1 2022, with only 3830 arrivals, compared to 6,155,269 in Q4 2018. Consequently, shopping tourism virtually vanished, with the number of same-day tourists plummeting to a mere 81 in July 2021, in comparison with 3,486,420 in Q1 2019. Moreover,

retail sales experienced an unprecedented decline of 35.0% year-on-year in Q1 2020. During the COVID-19 period, retail sales primarily relied on local residents' consumption, making it the dominant market segment.

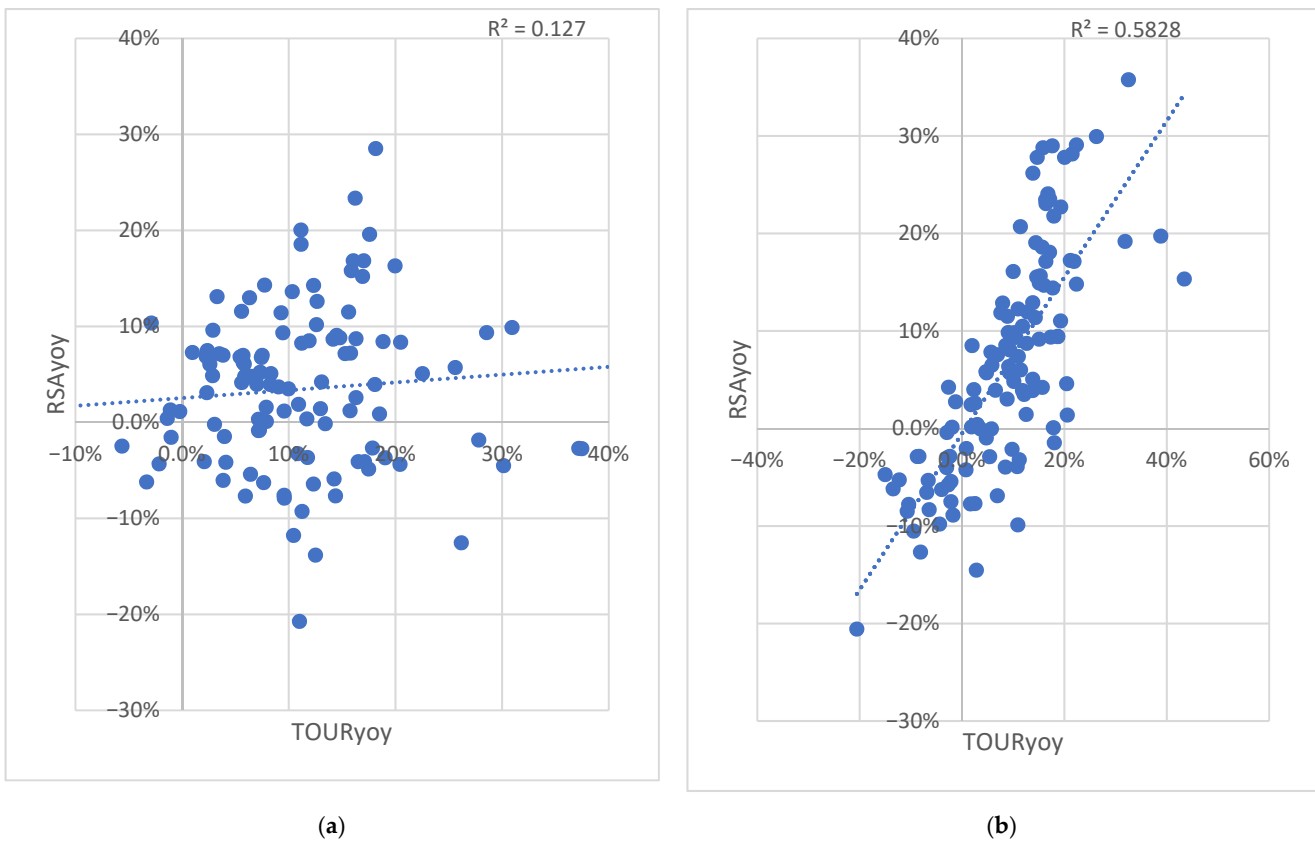

(**a**)                                          (**b**)

**Figure 1.** (**a**) Scatterplot of Year-on-Year Changes of Retail Sales Amount (RSAyoy) and No. of Tourists (TOURyoy) in 1999–2008. The positive correlation between the two series is weaker (Corr. Coeff. = 0.36), and the R-sq. is lower (0.127) in this pre-MEP period. Regression Line Equation: $R\hat{S}Ayoy = 0.0813(TO\hat{U}Ryoy) + 0.0254$. (**b**) Scatterplot of Year-on-Year Changes of Retail Sales Amount (RSAyoy) and No. of Tourists (TOURyoy) in 2009–2019. The positive correlation between the two series is stronger (Corr. Coeff. = 0.77), and the R-sq. is higher (0.583) in this MEP period. Regression Line Equation: $R\hat{S}Ayoy = 0.8011(TO\hat{U}Ryoy) - 0.0051$.

Figure 2 shows the strong correlation between the two series after the SARS epidemic in 2003. The correlation coefficient between the two series in the period of Q1 2000 to Q4 2018 is 96.3%.

Between Q1 2000 and Q4 2018, the number of tourists in Hong Kong saw a significant increase, rising from 0.99 million to over 6.16 million per quarter. However, a majority of this growth can be attributed to same-day visitors, indicating a strong surge in shopping tourism. In 2018, their collective expenditure on tourism reached HK$328.2 billion (US$42.1 billion) (HKTB 2023). Hong Kong was globally recognized as the most visited city that year (SCMP 2019), establishing itself as a prime destination for shopping tourists. On average, 56% of tourists in the two years leading up to the pandemic (2018–2019) were same-day visitors. Same-day travel became a distinguishing characteristic of cross-border shopping tourists in Hong Kong, allowing cross-border parallel traders to minimize their accommodation costs by completing their shopping within a single day.

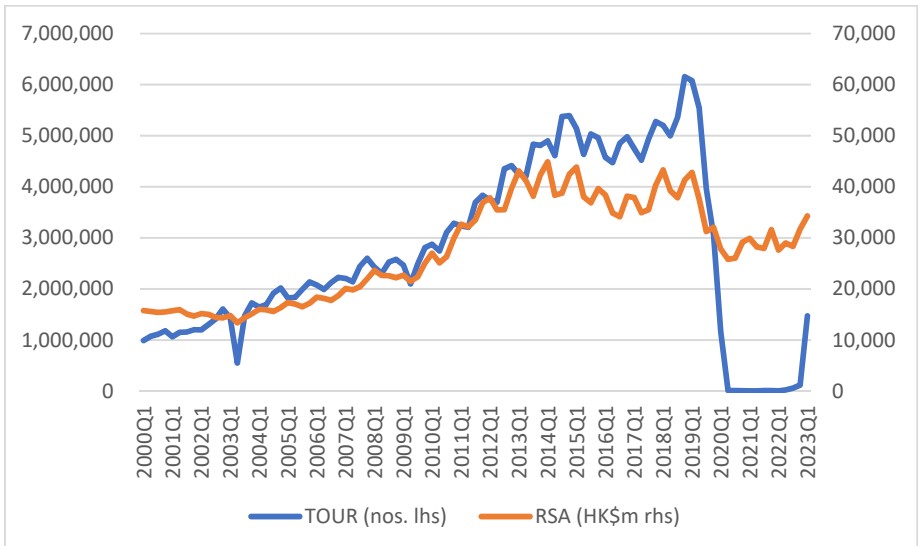

**Figure 2.** Number of tourist arrivals ($TOUR_q$) and retail sales amount ($RSA_q$) in Hong Kong, Q1 2000–Q1 2023. Sources: HK C&SD (2023b) and HK RVD (2023a).

The COVID-19 pandemic had a profound impact on shopping tourism in Hong Kong, causing significant disruptions to international and cross-border travel. Travel restrictions, quarantine measures, and diminished consumer confidence resulted in a sharp decrease in the number of shopping tourists visiting the city. The absence of mainland Chinese shopping tourists, who constituted a substantial portion of Hong Kong's shopping tourists, particularly affected the retail sector. This decline is evident in the drastic reduction in the ratio of same-day visitors to the overall number of visitors during the pandemic, dropping from a peak of 64% in January 2020 to a mere 0.3% in August 2021 (Figure 3). Hong Kong's tourism sector has experienced a rapid recovery since the reopening of borders in March 2023. The number of tourists has surged from 3830 in Q1 2022 to 1,471,584 in Q1 2023. Shopping tourism has also started to regain momentum, with the ratio of same-day tourists to the total number of tourists climbing back up to 47% in Q1 2023.

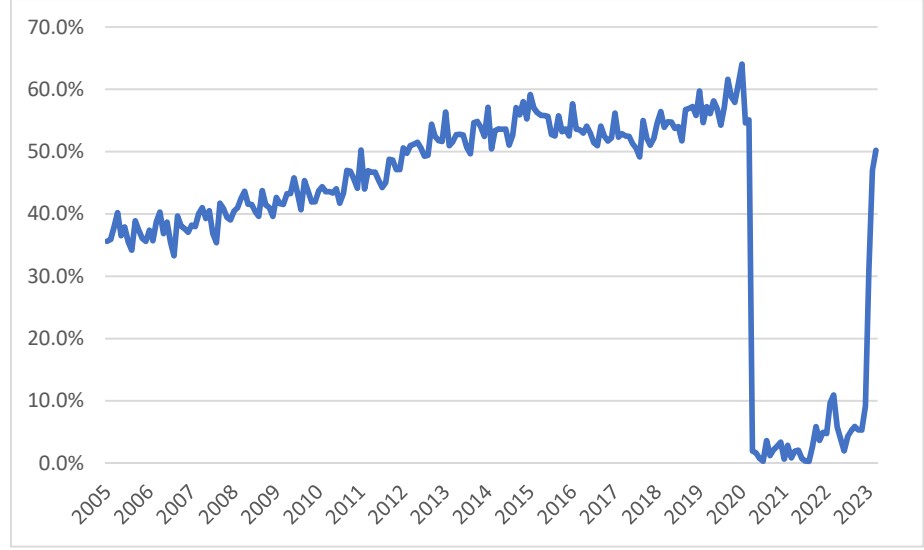

**Figure 3.** Proportion of same-day tourists to total number of tourist arrivals in Hong Kong, January 2005–March 2023. Source: HKTB (2023).

*1.2. Retail Sales and Rents in Hong Kong*

Figure 4 demonstrates a strong correlation between the change in retail rental index and the change in retail sales amount. Over the period from 2000 to 2022, the quarterly and yearly series exhibit correlation coefficients of 72.6% and 77.4%, respectively. Notably, in 2020, when retail sales experienced a decline of 24.3%, retail rents also fell by 9.2%.

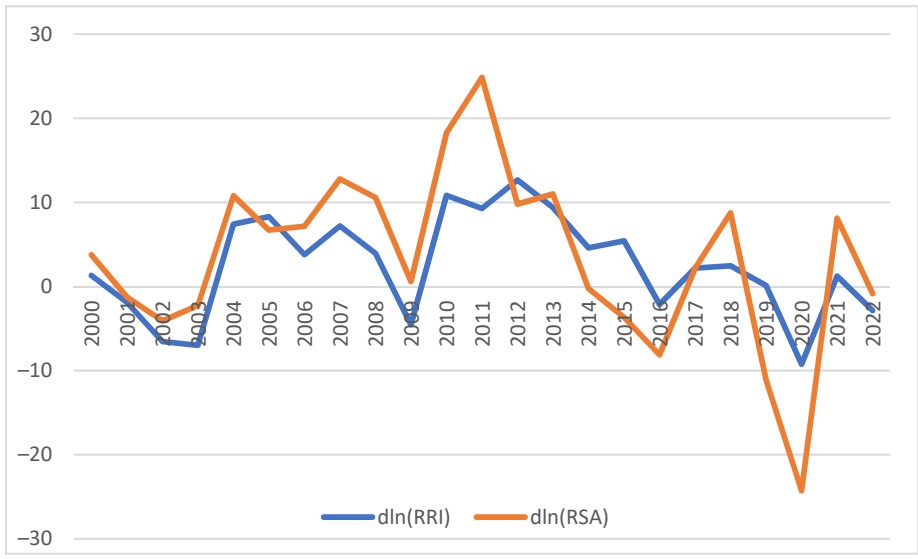

**Figure 4.** Year-on-year change of retail rental index ($dln(RRI_y)$) and year-on-year change of retail sales amount ($dln(RSA_y)$) in Hong Kong, 2020–2022. Sources: HK RVD (2023a) and HK C&SD (2023a).

This study aims to analyse the shopping tourism hypothesis, which suggests that in destinations popular among shopping tourists, such as Hong Kong, shopping tourism plays a significant role in shaping the demand for retail spaces, consequently impacting retail rents. The paper is structured as follows: Section 2 critically reviews the existing literature on the shopping tourism hypothesis. Section 3 outlines the research design, data sources, and methodology employed. The empirical results and robustness tests are presented in Section 4. Section 5 delves into the interpretation of the findings and discusses their implications, and Section 6 concludes the study.

## 2. Literature Review

### 2.1. The Determinants of Retail Rents

Traditionally, the fields of retailing and retail properties have been treated as separate disciplines. Real estate researchers have primarily focused on studying the determinants of retail rents, with little consideration given to tourism, particularly shopping tourism, as a significant factor. Early studies, such as that by Sirmans and Guidry (1993), explored determinants of shopping centre rents, primarily focusing on factors such as architectural design, location, and general economic conditions.

Other studies examined supply and demand factors in the context of retail space. For instance, Ke and White (2015) highlighted the importance of supply and demand dynamics in retail space. Tsolacos (1995) and D'Arcy et al. (1997) considered demand factors such as consumer spending and economic growth. Occasionally, consumer spending was further dissected into components like population growth and income levels (Kang 2019), without incorporating the impact of tourism shopping or shopping tourism into the analysis.

### 2.2. The Impacts of Tourism on Retail Markets

In recent years the impact of tourism on retail markets has attracted many retailing researchers' attention. Traditionally, shopping is only considered a secondary activity of a

trip, which is coined as "tourist shopping". It has been found to be a driver of retail sales in tourist destinations (Mehta et al. 2014; Silva and Hassani 2022).

However, the rise of "shopping tourism" has completely transformed the connection between tourism and the retail industry. Shopping tourism refers to a growing trend where shopping becomes the primary purpose of a trip (Timothy 2005; UNWTO 2014; Silva et al. 2019). It encompasses activities like cross-border shopping, parallel trade, and international out-shopping (Sharma et al. 2018). Extensive research has been conducted globally to understand the reasons behind cross-border shopping tourism, including for the borders between Canada and the US, various European countries, Malaysia and Singapore, Mainland China and Hong Kong, Germany and Denmark, Mexico and the US, and Thailand and Laos (Piron 2002; Baruca and Zolfagharian 2012; Lau et al. 2005; Nguyen et al. 2016; Makkonen 2016; Hadjimarcou et al. 2017; Boonchai and Freathy 2020). One common motivation for engaging in cross-border shopping tourism is the opportunity to benefit from differences in taxes and/or product quality (UNWTO 2014). However, there is a lack of research exploring the impacts of shopping tourism on retail property markets. This paper, therefore, takes Hong Kong as one of the shopping tourist destinations for studying the impacts of shopping tourism on retail sales and rents, given the large number of cross-border shopping tourists from Mainland China to Hong Kong.

Since the boom of shopping tourism, retail sales in shopping tourist destinations have experienced substantial growths. Research by Lee and Choi (2019) indicated that shopping tourists tend to spend more on shopping compared to other tourist segments. Lau et al. (2005) highlighted the importance of cross-border shopping in Hong Kong as a key factor in attracting shopping tourists. Liu and Wang (2010) also found a positive correlation between shopping tourism and destination attractiveness in Hong Kong.

Theoretically, retail spaces in tourist destinations in higher demand would lead to rising rental prices. Landlords could capitalize on the popularity of tourist areas by increasing rents and seeking tenants that cater to tourist spending. More recently, Li et al. (2018) and Liu et al. (2020) found a positive relationship between retail shops' prices and shopping tourism. However, on the one hand, they related tourism to retail property prices without controlling investment factors, and on the other hand, they did not control endogeneity biases, such as confounding factors and reversal causality issue. This study, therefore, analyses retail rents instead of prices and exploits the exogenous shocks on tourism during the pandemic period to analyse the impacts of shopping tourism on retail sales and rents. This quasi-experimental approach can establish the causality relationship between the variables.

The COVID-19 pandemic brought unprecedented disruptions to shopping tourism and its impacts on retail sales. Travel restrictions, lockdown measures, and health concerns significantly reduced tourist arrivals worldwide. Frago (2021) observed a sharp decline in the number of tourists and their spending capacity during the pandemic. This decline had a severe impact on retail sales, especially for businesses heavily reliant on shopping tourism.

The pandemic caused a downturn in the tourism industry, leading to a decrease in the demand for retail spaces. This, in turn, impacted rental prices. Property owners and landlords faced challenges in attracting tenants, which led to a decline in rental prices as a means to fill vacancies. Even though there were lockdowns and more e-Commerce during the pandemic (Nanda et al. 2021), their impacts lasted for a shorter period than border closure. This provides an exogenous shock to test shopping tourism effects on retail markets.

## 3. Materials and Methods

### 3.1. Data

All the data used in this paper come from three official sources: the Census and Statistics Department (HK C&SD) and the Rating and Valuation Department (HK RVD) of the Hong Kong SAR Government, and the Hong Kong Tourist Board (HKTB). Table 1 lists the descriptive statistics of the variables used in the analyses; their symbols, sources and

units of measure are shown in the notes to the table. Since the issues will be analysed using two sets of data series in different time frequencies, a subscript $t = y \text{ or } q$ in each variable is used to represent yearly or quarterly series.

**Table 1.** Descriptive Statistics of Variables, 2000–2022 and Q1 2000–Q4 2022.

| Variable | Yearly Time Series | | | | Quarterly Time Series | | | |
|---|---|---|---|---|---|---|---|---|
| | Mean | Std Dev | Min | Max | Mean | Std Dev | Min | Max |
| $dln(RRI_t)$ | 0.022 | 0.059 | −0.097 | 0.119 | 0.005 | 0.021 | −0.058 | 0.045 |
| $dln(RSA_t)$ | 0.029 | 0.104 | −0.278 | 0.222 | 0.008 | 0.070 | −0.186 | 0.128 |
| $dln(CPI_t)$ | 0.015 | 0.024 | −0.037 | 0.051 | 0.004 | 0.011 | −0.023 | 0.037 |
| $dln(STK_t)$ | 0.011 | 0.017 | −0.003 | 0.088 | | | | |
| $dln(VAC_t)$ | 0.017 | 0.110 | −0.199 | 0.277 | | | | |
| $dln(RGDP_t)$ | 0.028 | 0.038 | −0.068 | 0.083 | 0.007 | 0.056 | −0.130 | 0.114 |
| $dln(TOUR_t)$ | −0.127 | 1.056 | −3.665 | 1.889 | −0.024 | 0.603 | −4.866 | 1.725 |
| No. of Periods | 23 (2000–2022) | | | | 92 (Q1 2000–Q4 2022) | | | |
| $SDT_t/TOUR_t$ | 0.42 | 0.147 | 0.020 | 0.620 | 0.414 | 0.176 | 0.010 | 0.633 |
| No. of Periods | 23 (2000–2022) | | | | 72 (Q1 2005–Q4 2022) | | | |

Notes: RRI = Retail Rental Index, 1999 = 100; RSA = Retail Sales Amount, in HK\$M; CPI = Consumer Price Index, 2019M10–2020M9 = 100; STK = Stock of Private Commercial Properties at year end, in sq. m.; VAC = Vacant Private Commercial Properties at year end, in sq. m.; RGDP = Gross Domestic Product in chained (2021) dollars, in HK\$ m; TOUR = Tourist Arrivals, in numbers; SDT/TOUR = number of same-day tourists to total number of tourists ratio. Sources: RRI is collected from HK RVD (2023a), STK, VAC are from HK RVD (2023b), SDT is collected from HKTB (2023), but it is not available until 2005, and RSA, TOUR, CPI and RGDP are from HK C&SD (2023a, 2023b, 2023c, 2023d).

Furthermore, all the series are tested for their stationarity by two unit root tests. Table 2 shows the results of the stationarity tests by applying the Augmented Dickey–Fuller (ADF) and the Philips–Perron (PP) tests. Almost all variables in both yearly and quarterly time series are non-stationary in their level terms and stationary in their first differences in either test, except the yearly inflation variable $dln(CPI_t)$, which is marginally stationary in its first difference, with t-statistics = −2.33 ($p$-value = 0.17) and −2.36 ($p$-value = 0.16) in the ADF and PP tests.

**Table 2.** Unit Root Tests of Variables, 2000–2022 and Q1 2000–Q4 2022.

| Variable | Yearly Time Series (2000–2022) | | | | Quarterly Time Series (Q1 2000–Q4 2022) | | | |
|---|---|---|---|---|---|---|---|---|
| | Level | | First-Difference | | Level | | First-Difference | |
| | ADF | PP | ADF | PP | ADF | PP | ADF | PP |
| $ln(RRI_t)$ | −1.06 | −0.77 | −3.53 ** | −3.53 ** | | | | |
| $ln(RSA_t)$ | −2.15 | −1.30 | −3.33 ** | −3.31 ** | −1.25 | −1.29 | −2.81 * | −8.93 *** |
| $ln(CPI_t)$ | −0.55 | 0.47 | −2.33 | −2.36 | −0.25 | 1.32 | −2.62 * | −9.30 *** |
| $ln(STK_t)$ | −1.25 | −1.25 | −4.73 *** | −4.73 *** | | | | |
| $ln(VAC_t)$ | −1.26 | −1.26 | −4.86 *** | −4.87 *** | | | | |
| $ln(RGDP_t)$ | −2.46 | −2.55 | −3.96 *** | −3.96 *** | −2.12 | −1.39 | −1.52 | −13.00 *** |
| $ln(TOUR_t)$ | −1.45 | −0.89 | −8.67 *** | −3.36 ** | −3.54 *** | −1.67 | −8.33 *** | −8.40 *** |
| | Yearly time series (2000–2022) | | | | Quarterly time series (Q1 2005–Q4 2022) | | | |
| $SDT_t/TOUR_t$ | −1.62 | −1.62 | 0.44 | −4.95 *** | −1.32 | −8.78 *** | −1.30 | −8.80 *** |

Notes: Figures are t-statistics; ***, **, and * represent $p$-values $\leq$ 0.01, 0.05, and 0.10, respectively. ADF and PP tests are the Augmented Dickey–Fuller Unit Root Test and the Philips–Perron Unit Root Test, respectively. The longest available quarterly data series for the number of same-day tourists SDT starts from Q1 2005.

### 3.2. Research Design

Before the outbreak of COVID-19, Hong Kong was one of the most popular shopping tourism destinations, with more than 6 million tourists per quarter. The period was marked by a continually growing trend of retail rents in Hong Kong. There are two different explanations for this phenomenon. One concerns landlord hegemony and the escalation of retail rents, which caused retailers to raise the prices of retail goods, thus resulting in higher retail sales amounts. The other is that shopping tourism triggered higher retail sales and caused higher retail rents. Neither hypothesis has empirical evidence to support it. In fact, a Granger Causality test on the series could not determine the direction of causality, as both retail sales and retail rents demonstrated Granger Cause in the monthly and quarterly series and both real retail sales and tourist arrivals demonstrated Granger Cause in the yearly series, as shown in Table 3. A quasi-experiment is required to introduce an external shock as a treatment on some variables to identify the causal links.

This study leverages the COVID-19 period from 2020 to 2022 as a natural quasi-experiment to investigate the causal relationship between tourist arrivals, retail sales, and retail rents. The onset of the pandemic in 2020 led to a near-total shutdown of borders, which can be viewed as an external shock to the tourism industry. As a consequence, the number of tourist arrivals experienced a drastic decline, nearing zero. With the closure of borders, retail sales became largely dependent on local residents' consumption. The significant drop in both retail sales and rents can be primarily attributed to the decline in shopping tourism activities.

The causal link between retail sales and retail rents is well-acknowledged in both academic theory and market practices, particularly through turnover rent mechanisms (Cheung and Yiu 2022). Model 1 examines this causal link by controlling for supply-side factors and inflation. As the supply-side data are available on a yearly basis, Model 1 operates on a yearly basis as well.

To explore the causal links between tourist arrivals, retail sales, and rents, three models using quarterly data series are employed. Model 2 serves as the baseline model, analysing the overall impact of tourist arrivals on retail sales while keeping inflation and real economic growth constant. Models 3 and 4 utilize a differences-in-differences (DID) approach to identify the moderating effects of the MEP scheme and the COVID-19 pandemic on the relationship between tourist arrivals and retail sales, with all else being equal. Lastly, Model 5 integrates the aforementioned models to investigate the causal link from tourist arrivals to retail rents in the yearly series, while controlling for other factors.

**Table 3.** Granger Causality Tests of Variables, 2000–2022 and 2000Q1–2022Q4.

| Variable | Yearly Time Series (2000–2022) | | Quarterly Time Series (2000Q1–2022Q4) | | Monthly Time Series (2000M01–2022M12) | |
|---|---|---|---|---|---|---|
| | Obs | F-Stat | Obs | F-Stat | Obs | F-Stat |
| $dln(RRI_t)$ Granger Cause $dln(RSA_t)$ | 23 | 0.35 | 92 | 8.71 *** | 276 | 5.97 *** |
| $dln(RSA_t)$ Granger Cause $dln(RRI_t)$ | 23 | 1.03 | 92 | 4.15 ** | 276 | 3.42 ** |
| $dln(RSA_t/CPI_t)$ Granger Cause $dln(TOUR_t)$ | 23 | 3.02 * | 92 | 0.08 | 276 | 0.98 |
| $dln(TOUR_t)$ Granger Cause $dln(RSA_t/CPI_t)$ | 23 | 2.68 * | 92 | 0.29 | 276 | 0.60 |

Notes: Figures are F-statistics; ***, **, and * represent *p*-values $\leq$ 0.01, 0.05, and 0.10, respectively. F-statistics are estimated by 2-lag pairwise Granger Causality test. Retail sales amount RSA is divided by consumer price index CPI to control the inflation factor.

### 3.3. Empirical Models of the Quasi-Experiment

In this study, five time series regression models are examined, Model 1 tests the first hypothesis of the effect of retail sales on retail rents (Equation (1)). The dependent variable is the growth rates of retail rental index $dln(RRI_t)$. The retail rental index can be explained by supply and demand factors and inflation $dln(CPI_t)$. Supply factors are proxied by the growth rates of commercial property stocks $dln(STK_t)$ and that of vacant commercial

properties $dln(VAC_t)$. The demand factor is proxied by the growth rate of retail sales amount $dln(RSA_t)$. A first-order autoregressive AR(1) model is applied. Due to data limitations, supply factors can only be studied in yearly series.

$$(1 - \rho_1 L)[dln(RRI_t) \quad -(c_1 + \beta_1 dln(CPI_t) + \beta_2 dln(RSA_t) + \gamma_1 dln(STK_t) + \gamma_2 dln(VAC_t))] \\ = \varepsilon_{1,t} \ldots \ldots \tag{1}$$

where $\rho_i$ is the first-order serial correlation coefficient and $L$ is a lag operator. $CPI_t$, $RSA_t$, $STK_t$, and $VAC_t$ are consumer price index, retail sales amount, commercial property stock, and vacant commercial properties in Hong Kong at time t, respectively. $dln$ represents the growth rates of the variables by taking the first difference of the natural logarithm of the variables. $c_i, \beta_i, \gamma_i$ are coefficients to be estimated. $\varepsilon_{i,t}$ are the error terms.

Model 2 tests the second hypothesis of the effect of tourist arrivals on retail sales. The dependent variable is the growth rates of retail sales amount $dln(RSA_t)$, and the explanatory variables include inflation rate $dln(CPI_t)$, economic real growth rate $dln(RGDP_t)$, and growth rate of tourist arrivals $dln(TOUR_t)$ (Equation (2)). It can be studied using quarterly series.

$$(1 - \rho_2 L)[dln(RSA_t) \quad -(c_2 + \beta_3 dln(CPI_t) + \beta_4 dln(RGDP_t) + \gamma_3 dln(TOUR_t))] \\ = \varepsilon_{2,t} \ldots \ldots \tag{2}$$

where $RGDP_t$, and $TOUR_t$ are real GDP and number of tourist arrivals in Hong Kong at time $t$, respectively.

Models 3 and 4 apply a DID approach to test the moderating effects of the MEP scheme (i.e., in 2009–2019, Equation (3)) and the Covid pandemic (i.e., in 2020–2022, Equation (4)) on the impacts of tourist arrivals on retail sales.

$$(1 - \rho_3 L)[dln(RSA_t) \quad -(c_2 + \beta_3 dln(CPI_t) + \beta_4 dln(RGDP_t) + \gamma_3 dln(TOUR_t) \\ +\gamma_4 MEP_t \times dln(TOUR_t) + \gamma_5 MEP_t)] = \varepsilon_{3,t} \ldots \ldots \tag{3}$$

$$(1 - \rho_4 L)[dln(RSA_t) \quad -(c_2 + \beta_3 dln(CPI_t) + \beta_4 dln(RGDP_t) + \gamma_3 dln(TOUR_t) \\ +\gamma_6 Covid_t \times dln(TOUR_t) + \gamma_7 Covid_t)] = \varepsilon_{4,t} \ldots \ldots \tag{4}$$

where $MEP_t = 1$ when the period is 2009–2019, and $Covid_t = 1$ when the period is 2020–2022.

Model 5 is a combined regression model that combines Equations (1) and (3) to test the effect of tourist arrivals on retail rents in the yearly series. All the regression models are estimated using EViews (2022) software.

## 4. Results

Table 4 shows the results of Models 1 to 6. In Model 1, the results support the demand factor that retail sales and inflation impose positive impacts on retail rents, but not the supply factors. Model 2 establishes the positive association between tourist arrivals and retail sales amount, after controlling inflation and economic growth factors. Yet, as shown in Models 3 and 4, the effect was much stronger after the MEP scheme and before the pandemic, i.e., in the period of 2009–2019, implying that the effect was mainly derived from shopping tourism. The effect of number of tourists on retail sales increased by more than threefold during this period. This shows that a 1% increase in the number of tourists led to an increase of 0.49% in retail sales in the MEP period, in comparison with just 0.12% before 2009. However, the COVID-19 pandemic brought the effect of number of tourists on retail sales back to an unprecedentedly low magnitude, at just about 0.02% (0.37–0.35).

Furthermore, Model 5 is the combined regression model in the yearly series. The combined results also confirm the shopping tourism hypothesis that a 1% increase per annum in tourist arrivals before the pandemic was associated with a 0.23% increase per annum in retail rents. The effect almost vanished during the COVID-19 period. The explanatory power of the model is reasonably high, at almost 87% (Adj R-squared).

**Table 4.** Results of the regression models.

| Models | Model 1 Supply–Demand | Model 2 Tourist and Sales | Model 3 MEP and Sales | Model 4 COVID and Sales | Model 5 Combined Yearly | Model 6 Robustness Test |
|---|---|---|---|---|---|---|
| Dependent Variables | $dln(RRI_y)$ | $dln(RSA_q)$ | $dln(RSA_q)$ | $dln(RSA_q)$ | $dln(RRI_y)$ | $dln(RSA_q)$ |
| Constant | 0.001 (0.07) | −0.001 (−0.19) | 0.011 (0.84) | −0.023 (−2.42) ** | −0.025 (−1.94) * | 0.007 (0.33) |
| $dln(CPI_t)$ | 1.002 (2.87) ** | 2.307 (3.20) *** | 2.938 (4.64) *** | 3.68 (3.49) *** | 1.465 (10.01) *** | 3.174 (2.95) *** |
| $dln(RSA_t)$ | 0.288 (2.88) ** | | | | | |
| $dln(STK_t)$ | 0.023 (0.02) | | | | −0.093 (−0.15) | |
| $dln(VAC_t)$ | −0.138 (−1.50) | | | | −0.125 (−1.33) | |
| $dln(RGDP_t)$ | | 0.130 (0.81) | −0.303 (−1.38) | 0.213 (1.05) | 0.509 (2.27) ** | 0.232 (0.91) |
| $dln(TOUR_t)$ | | 0.031 (2.70) *** | 0.117 (1.83) * | 0.375 (5.45) *** | 0.225 (2.18) ** | 0.014 (0.42) |
| $dln(TOUR_t) * MEP_t$ | | | 0.370 (4.59) *** | | | |
| $MEP_t$ | | | −0.025 (−1.68) * | | | |
| $dln(TOUR_t) * Covid_t$ | | | | −0.354 (−4.73) *** | −0.221 (−2.09) * | |
| $Covid_t$ | | | | 0.020 (1.06) | −0.011 (−0.56) | |
| $(SDT_t/TOUR_t) \times dln(TOUR_t)$ | | | | | | 0.236 (2.36) ** |
| $SDT_t/TOUR_t$ | | | | | | −0.047 (−1.02) |
| $AR(1)$ | 0.111 (0.52) | −0.033 (−0.26) | −0.026 (−0.12) | −0.204 (−1.09) | −0.394 (−1.54) | −0.113 (−0.59) |
| $SIGMASQ$ | 0.001 (2.14) ** | 0.004 (5.75) *** | 0.002 (5.46) *** | 0.004 (3.61) *** | 0.0003 (1.54) | 0.004 (3.74) *** |
| No. of Observations | 23 (2000–2022) | 92 (2000Q1–2022Q4) | 80 (2000Q1–2019Q4) | 56 (2009Q1–2022Q4) | 23 (2000–2022) | 56 (2009Q1–2022Q4) |
| Adj. R-sq | 0.685 | 0.172 | 0.393 | 0.395 | 0.870 | 0.316 |

Notes: Figures in parenthesis are t-statistics; ***, **, and * represent *p*-values $\leq$ 0.01, 0.05, and 0.10, respectively. Coefficients are estimated by the ARMA Maximum Likelihood (OPG-BHHH) Method, with coefficient covariance computed using the outer product of gradients. Legends: RRI = Retail Rental Index; RSA = Retail Sales Amount; CPI = Consumer Price Index; STK = Stock of Private Commercial Properties; VAC = Vacant Private Commercial Properties; RGDP = Real Gross Domestic Product; TOUR = Tourist Arrivals; SDT = number of same-day tourists; MEP = multiple-entry permit year—dummy; COVID = COVID-19 pandemic year—dummy.

As a robustness test on the shopping tourism hypothesis, Model 6 is a DID model on the moderating effect of the same-day to total tourists ratio $SDT_t/TOUR_t$ on the tourist impact on retail sales. The results of the quarterly series confirm the positive moderating effect of $SDT_t/TOUR_t$, implying that the weaker effect of number of tourists on retail sales amount in the COVID period found in Model 4 is due to the sharp reduction in the same-day tourists, who are mostly cross-border shopping tourists.

## 5. Discussion

There have been some studies on the impact of tourist arrivals on retail sales, but very few of them have empirically analysed the effects of shopping tourism on retail sales. In addition, there have been no studies connecting the link between tourism and retail sales to the link between retail sales and rents. Some studies have aimed to directly measure

the impact of tourism on the prices of retail shops; however, it is difficult to control for the many investment factors.

In fact, it is quite difficult to disentangle the effect of shopping tourism on retail sales from local residents' spending because, on the one hand, data on shopping tourists are not available, and on the other hand, the actual amount of tourist spending is difficult to determine. This study makes use of the period of MEP and the COVID-19 pandemic in Hong Kong to identify the emergence of shopping tourism after the implementation of the MEP and the suspension of shopping tourism during the pandemic period. This study divides the time series into three sub-periods to test the hypothesis: (1) the pre-MEP period, 2000–2008; (2) the MEP period, 2009–2019; and (3) the COVID period, 2020–2022. This enables time series analyses of the impacts of shopping tourists on retail sales and rents. Due to the data frequency, two levels of regression analysis are carried out: retail rental analysis is conducted on a yearly basis, whereas retail sales analysis is determined on a quarterly basis. The results confirm that the number of tourists is one of the major determinants of retail sales amounts and retail rents in Hong Kong. However, the effect of the number of tourists on retail sales was weak before cross-border shopping tourism became commonplace, i.e., in the pre-MEP period, and after the outbreak of the pandemic, i.e., in the COVID period. Shopping tourism is found to increase the impacts of tourist arrivals on retail sales by more than three times. Furthermore, a robustness test on the hypothesis is conducted using the number of same-day tourists as a proxy of the number of shopping tourists.

The COVID-19 pandemic served as a quasi-experiment that unveiled the intricate causal relationship between shopping tourism, retail sales, and retail rents. While the pandemic's disruption had detrimental effects on the retail sector, it also presented valuable lessons on urban governance and risk management (Blake and Sinclair 2003). Shopping tourism may bring good business to retailers, but it could also cause high volatility and high retail rents (Cheung and Yiu 2022). By leveraging these insights, retailers can build resilience, attract customers, and thrive even in uncertain times, reducing their dependence on shopping tourism and ensuring a more sustainable future.

## 6. Conclusions

This novel study uses the COVID-19 pandemic as a quasi-experiment to investigate the impact of shopping tourism on retail sales and rents in Hong Kong. Shopping tourism refers to individuals who travel primarily for shopping purposes, and their spending patterns can have significant effects on the retail sector. The COVID-19 pandemic disrupted global travel and resulted in a decline in shopping tourist arrivals, leading to a downturn in sales for retailers, especially in popular shopping tourism destinations. The reduced demand for retail spaces is reflected by a reduction in rentals. The research analyses the relationship between tourist arrivals, retail sales, and rents using time series analysis and identifies the impact of shopping tourism on retail rents. The results show a positive association between tourist arrivals and retail sales and rents during the period of shopping tourism growth before the pandemic.

A practical implication of this study is the aim of retail resilience through customer diversification. The COVID-19 pandemic underscored the importance of diversifying the customer base for retailers in tourist destinations. Overreliance on shopping tourism left businesses vulnerable to external shocks, emphasizing the need for a more balanced approach that caters to both locals and tourists.

Future research should further investigate the long-term impacts of shopping tourism on retail sales and retail rents after the pandemic. Comparative studies between destinations that heavily rely on shopping tourism and those with a diversified customer base would provide valuable insights into the best strategies for building resilient retail sectors. As emphasized in the UNEP & UNWTO's (2005) Sustainable Tourism Development guidelines and management practices, respecting the socio-cultural authenticity of host communities and contributing to inter-cultural understanding and tolerance is one of the core values of

sustainable tourism (Su et al. 2017). Resident-friendly policies on tourism are conducive to achievement of the Sustainable Development Goals (Zhang et al. 2019).

**Funding:** This research received no external funding.

**Institutional Review Board Statement:** Not applicable.

**Informed Consent Statement:** Not applicable.

**Data Availability Statement:** Data are available in a publicly accessible repository that does not issue DOIs. Publicly available datasets were analysed in this study. This data can be found in the following reference sources: RRI: retail rental index (1999 = 100), from HK RVD (2023a); RSA: retail sales amount (in HK$M), from HK C&SD (2023a); TOUR: tourist arrivals (in numbers), from HK C&SD (2023b); CPI: consumer price index (2019M10–2020M9 = 100), from HK C&SD (2023c); RGDP: gross domestic products (in chained 2021 dollars, HK$M), from HK C&SD (2023d); RRI: retail rental index (1999 = 100), from HK RVD (2023a); STK: stock of private commercial properties at year end (in square meters), from HK RVD (2023b); VAC: vacant private commercial properties at year end (in square meters), from HK RVD (2023b); SDT: number of same-day tourists (in numbers), from HKTB (2023).

**Conflicts of Interest:** The author declares no conflict of interest.

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
