# Peer review of "The Impacts of Shopping Tourism on Retail Sales and Rents: Lessons from the COVID-19 Quasi-Experiment of Hong Kong"

_jrfm, doi:10.3390/jrfm16060301_

Round 1
Reviewer 1 Report
It was agreeable to read your manuscript, as it describes clearly the situation in HK for retails sales, tourism and retail rents. The reader gets a broad view of the situation. It was a pity that the time period is quite small, but, as you point out, the measures to facilitate crossing from the mainland limit still more the real time span.
I would suggest than when you present a regression line, include de error term in it, or indicate that it is Å· instead of y.
About Granger tests to study the direction of causality, surely with 23 yearly data (less, when differencing), it is not easy to obtain clear results as you point; it gets better with quarterly data. Using the closing of borders has been an opportunity to isolate the shopping by locals from tourists.
Reviewer 2 Report
Thanks for the opportunity to review this article and congratulations to the author for his research. The research shows us the impact the COVID-19 pandemic had on shopping tourism on retail sales and rents in Hong Kong. It analyzes the relationship between tourist arrivals, retail sales and rents based on time series and identifies the impact of shopping tourism on retail space and rents.
The introduction is well written, shows the dynamics of tourist numbers in the pre-pandemic and pandemic period, links to where the analysed data was used are required.
The bibliography used is adequate for the study, but could be extended with more recent studies.
The methodology used is adequate for the research, presenting a number of 5 econometric models, correctly written, it is necessary to write the links where the analyzed data were used, as well as the analysis software used for the calculation of the indicators.
The results obtained are in line with the research carried out, I suggest writing the discussion chapter separately from the conclusions.
